# Analysis of the mechanism of *Ricinus communis* L. tolerance to Cd metal based on proteomics and metabolomics

**Zhao Huibo**[1], **Zhao Yong**[1,2], **Luo Rui**[1], **Li Guorui**[1], **Di Jianjun**[1], **Wen Qi**[1], **Liang Xiaotian**[1], **Yin Mingda**[1], **Wen Yanpeng**[1], **Wang Zhiyan**[1], **Huang Fenglan**[1,3,4,5,6,7]*

**1** School of Life Science and Food, Inner Mongolia University for Nationalities, Tongliao, Inner Mongolia, China, **2** College of Life Sciences, Baicheng Normal University, Baicheng, Jilin, 137099, China, **3** Key Laboratory of Castor Breeding of the State Ethnic Affairs Commission, Tongliao, Inner Mongolia, China, **4** Inner Mongolia Key Laboratory of Castor Breeding, Tongliao, Inner Mongolia, China, **5** Inner Mongolia Engineering Research Center of Industrial Technology Innovation of Castor, Tongliao, Inner Mongolia, China, **6** Inner Mongolia Industrial Engineering Research Center of Universities for Castor, Tongliao, Inner Mongolia, China, **7** Inner Mongolia Collaborative Innovation Center for Castor Industry, Tongliao, Inner Mongolia, China

* huangfenglan@imun.edu.cn

**Data Availability Statement:** All relevant data are within the paper and its Supporting Information files.

## Abstract

The pollution of soil with heavy metals is an increasingly serious worldwide problem, and cadmium (Cd) has attracted attention because of its high toxicity to almost all plants. Since castor tolerates the accumulation of heavy metals, it has the potential for heavy metal soil remediation. We studied the mechanism of the tolerance of castor to Cd stress treatments at three doses: 300 mg/L, 700 mg/L, and 1,000 mg/L. This research provides new ideas for revealing the defense and detoxification mechanisms of Cd-stressed castor. By combining the results of physiology, differential proteomics and comparative metabolomics, we conducted a comprehensive analysis of the networks that regulate the response of castor to Cd stress. The physiological results mainly emphasize the super-sensitive responses of castor plant roots to Cd stress and the effects of Cd stress on plants' antioxidant system, ATP synthesis and ion homeostasis. We confirmed these results at the protein and metabolite levels. In addition, proteomics and metabolomics indicated that under Cd stress, the expressions of proteins involved in defense and detoxification, energy metabolism and other metabolites such as organic acids and flavonoids were significantly up-regulated. At the same time, proteomics and metabolomics also show that castor plants mainly block the root system's absorption of $Cd^{2+}$ by enhancing the strength of the cell wall, and inducing programmed cell death in response to the three different doses of Cd stress. In addition, the plasma membrane ATPase encoding gene (*RcHA4*), which was significantly upregulated in our differential proteomics and RT-qPCR studies, was transgenically overexpressed in wild type *Arabidopsis thaliana* for functional verification. The results indicated that this gene plays an important role in improving plant Cd tolerance.

**Funding:** The National Natural Science Foundation of China (31860071); New Agricultural Science Research and Reform Project of Ministry of Education (2020114); General project of Inner Mongolia Natural Science Foundation (2021MS03008); Inner Mongolia Grassland Talents Innovation Team -- Castor bean Molecular Breeding Research Innovation Team Rolling Support Program (2022); 2021 Higher Education Teaching Reform Research Project of the State Ethnic Affairs Commission (21082); Project of Basic Scientific Research Funds of Universities directly under the Autonomous Region in 2022 (237); Inner Mongolia Autonomous Region Castor Industry Collaborative Innovation Center Open Fund Project (MDK2021011, MDK2022014). Inner Mongolia University for Nationalities 2023 Basic Research Funds of Colleges and Universities directly under the Autonomous Region (225, 227, 244) The funders had no role in study design, data collection and analysis, decision to publish, or preparation of the manuscript.

**Competing interests:** The authors have declared that no competing interests exist.

## Introduction

With the continuous increase in the demand for steel, ore, etc., the excessive use of chemical fertilizers, and the rapid progress of urbanization, ecosystems have been severely damaged. The result of these excessive disturbances of ecosystems is not only reflected in the destruction of the availability of natural resources, but it also causes extensive and serious damage to water sources, soil and other necessary conditions for the survival of organisms [1–3]. Among pollutants, heavy metals not only damage soil structure, but also affect soil organic matter content and water holding capacity [4]. After excessive heavy metals are absorbed by plants, there are toxic reactions such as growth retardation, leaf crimping, degreening and yellowing. This is also an important reason for poor crop quality and low yield [5–7]. There are many kinds of metals in soil, including not only iron (Fe), zinc (Zn), copper (Cu), manganese (Mn) and other essential micronutrients for plant growth, but also cadmium (Cd), arsenic (As), mercury (Hg), lead (Pb), chromium (Cr) and other non-essential elements for plant growth. Among these metals, Cd is more easily absorbed and accumulated by plants than the others due to its high mobility, and may threaten human health by accumulating in the food chain. This makes soil cadmium pollution of great concern to people [8].

Long-term plant growth under Cd stress produces a variety of injuries, including the rapid generation of $O^{2-}$, $H_2O_2$, -OH and other reactive oxygen species (ROS). This destroys the intracellular oxidation system [9]; and damages photoreaction and carbon assimilation in plant photosynthesis. This reduces the photosynthetic rate of plants [10–14]. In addition, $Cd^{2+}$ impacts the ionic homeostasis of plants. In addition to inhibiting the absorption of $Ca^{2+}$, it may also interfere with the normal switching of channel proteins by inhibiting the current homeostasis of $K^+$ channels, thereby affecting plant growth [15].

Castor (*Ricinus communis* L.), as a non-edible oil crop, is widely cultivated and used because it is rich in many useful compounds [16]. Biodiesel and bio-lubricants synthesized from castor beans are widely used in the aerospace industry, also known as renewable "petroleum resources" [17–19]. Ricinine, an alkaloid which exists in stems, leaves and inflorescences, is used as a biological toxin that can inhibit or damage a variety of insects, so it is called a natural biological pesticide [20,21]. Meanwhile, ricin, a protein extracted from castor bean seeds, has become the focus of anti-cancer drug research at present. It acts mainly by inhibiting the 60S subunit of the ribosome; this inhibition of protein synthesis leads to cell death [22,23]. In addition, because of its ability to grow in poor and high salinity soils, producing stout stems and well developed roots and high biomass, it is known as the ideal green plant [24,25]. Non-edible castor is known for its heavy metal tolerance, especially for its super-enrichment and remediation capabilities for Pb, Zn, Cd and other heavy metals [26]. Therefore, it is called a heavy metal enrichment plant, and is widely used in the remediation of heavy metal contaminated soil [27,28].

With the development of transcriptomics, differential proteomics, comparative metabolomics and other monomic studies on plant responses to heavy metal Cd stress, it may become the latest trend in the field of omics to integrate multiple omics approaches to analyze plant responses to Cd stress. By understanding genes' expression and protein products, and analyzing their networks and metabolic pathways, it is possible to study the functional areas of the genome involved in the Cd stress responses of plants, that is, to understand the tolerance mechanisms and cell detoxification mechanisms of plants to Cd and other heavy metals from an overall perspective [29,30]. By combining conventional physiological research with the joint analysis of differential proteomics and differential metabolomics of the castor bean root system, this study revealed the effects of different concentrations of Cd on the growth and development of castor bean and the molecular mechanisms of castor bean's tolerance to heavy

metal Cd from an overall perspective. Castor genes that were upregulated by Cd stress were examined to identify crucial Cd detoxification genes. A castor plasma membrane ATPase was heterologously overexpressed in Arabidopsis to verify its role in Cd detoxification. This study creates a good foundation for revealing the mechanisms by which castor sequesters Cd and contributes to the remediation of Cd pollution in soil.

## Materials and methods

### Plant material and processing

"2129" castor bean strain was selected as the material for this study. Castor bean seeds were planted in plastic pots (50 cm\*20 cm\*14 cm) with white sand (9500 cm$^3$). After 14 days of normal culture and growth under 16/8 h light and dark cycle and 25/20˚C day/night cycle, the seedlings were randomly assigned to treatments with 0 mg/L, 300 mg/L, 700 mg/L, and 1,000 mg/L Cd solution (CdCl$_2$·5H$_2$O) for 21 days (plants were watered every 7 days with 1200ml of each treatment solution). The roots, stems and leaves of the treated plants were washed with distilled water and dried, then frozen in liquid nitrogen for subsequent research and analysis, and the treatments of 0, 300, 700 and 1,000mg/L were recorded as CK, ZA, ZB and ZC, respectively. Each sample consisted of three biological replicates, and each replicate consisted of mixed tissue from three castor plants. Note: The sand used in this experiment is standard 1mm dust-free white sand (white solid, loose soil, insoluble in water, good water and air permeability), and the heavy metal Cd pollutant generated during the study were treated and recycled in accordance with the requirements of public health institutions and environmental protection departments.

### Physiological index measurement

The determination of Cd content of plant tissues was completed by Qingdao Kechuang Quality Inspection Co., Ltd., and physiological indicators were determined with a physiological index determination kit (Keming Biotechnology Co., Ltd., China).

### Proteomics

After extracting total proteins (CK, ZA, ZB, ZC) from the roots of castor plant by the phenol extraction method, then, the peptides after trypsinization were labeled using the iTRAQ kit (AB SCIEX, USA)., and the labeled peptides were loaded onto the Agilent 300Extend C18 column (5 μm particle diameter, 4.6 mm inner diameter, 250 mm length), dissolved in the liquid chromatography mobile phase A (2% acetonitrile; 0.1% formic acid) and then passed into the ultra-high performance liquid system for separation. Liquid gradient setting: 5% ~ 35% mobile phase B (98% acetonitrile; 0.1% formic acid), 0 ~ 60 min, the flow rate is 300 nL/min. The peptides separated by the ultra-high-performance liquid system were injected into the NSI ion source for ionization, and then entered the Q Exactive HF mass spectrometer for analysis. The ion source voltage was set to 2.0 kV.

The proteins differentially expressed under the various Cd treatments that were identified by mass spectrometry were identified in the castor protein database, which was derived from the UniProt castor database. The raw format file that stores the complete mass spectrum scan data was imported directly into the Proteome Discoverer 2.2 software for database search, peptide spectroscopy, and protein quantification. The relative quantitative value of each protein in two comparative samples was tested by T-test, and a p-value $\leq$ 0.05 was considered significant.

## Metabolomics

The plant root material (the same set of samples as the proteomics analysis) was vacuum freeze-dried and ground to a powder, dissolved in 70% methanol and extracted at 4°C overnight. Samples were filtered through a microporous filter (0.22 μm pore size) and analyzed by Ultra performance liquid chromatography and tandem mass spectrometry (UPLC-MS/MS).

Based on the OPLS-DA results, a multivariate analysis of variable importance in project (VIP) of the OPLS-DA model was performed, Significant metabolites were identified with log 2 (fold change) $\geq$2 and VIP$\geq$1. The software Analyst 1.6.1 was used to analyze and process the mass spectrum data, and the R program (www.r-project.org/) was used to carry out normalized processing and hierarchical cluster analysis (HCA) for the identified metabolites that differed between the Cd treatments.

## RT—qPCR

The total RNA extraction kit (ZOMANBIO, Beijing, China) was used to extract RNA from the experimental materials, and 3μg RNA was used to synthesize the first strand cDNA in a 20μl reaction using the Reverse Transcription System kit (Promega, Beijing, China). The GoTaq® qPCR and RT-qPCR Systems kit (Promega, Beijing, China) was used to quantify the transcript abundance on the ABI 7500 real-time PCR system (Applied Biosystems). The genes were normalized with castor actin mRNA as the internal reference. The relevant primer sequences are shown in S1 Table. There were 3 biological replicates for each sample, the average CT value was used, and the RT-qPCR result was calculated by the $2^{-\Delta\Delta C}$ method.

## Subcellular localization

In order to reveal the function of a castor Cd stress-related gene, a gene that was significantly up-regulated in response to Cd treatment, based on proteomics and qRT-PCR analysis, was selected for functional verification. By querying its protein sequence with blastp at the National Center for Biotechnology Information (NCBI), we found that it encodes a plasma membrane ATPase protein, named *RcHA4* in this study.

In order to study the tissue and cell specificity of *RcHA4* (LOC8286312) gene expression, the CDS sequence of *RcHA4* was amplified and cloned into the pBI121-GFP vector, and onion epidermis was co-transformed with pBI121-*RcHA 4*-GFP to produce a fusion expression protein.

## Heterologous expression of *RcHA4* in *Arabidopsis*

The CDS sequence of the gene *RcHA4* was cloned into the pCAMBIA1305.2 vector with the CaMV35s promoter, and the recombinant expression vector pCAMBIA1305.2-*RcHA4* was transformed into wild-type *Arabidopsis* (Col-0 line) by the flower dipping method. The plants were screened with 50 mg/mL hygromycin on MS medium containing 2% sucrose and 0.8% agar (pH 7.0). The primers used to amplify the *RcHA4* gene is shown in S2 Table.

We analyzed the change in *RcHA4* gene expression in *Arabidopsis RcHA4*-OE plants under Cd stress. The method was as follows: After overexpressing *Arabidopsis thaliana* positive plants *RcHA4*-OE were cultured normally for 18 days, the positive plants without Cd treatment were taken as controls, and the *RcHA4*-OE plants were grown for 12 hours in medium containing 50 uM and 500 uM Cd solutions.Using RT-qPCR, the overexpressing plants' (*RcHA4*-OE) response to Cd stress was analyzed. In this experiment, *atACTIN* was used as the internal reference gene for normalization. The primer sequences of PCR and RT-qPCR are shown in S2 Table.

### Transgenic *Arabidopsis* Cd tolerance analysis

The seeds were disinfected with 75% alcohol for 10 min and then planted on MS solid medium. After vernalization at 4°C for 72 h, the seeds were moved to a constant temperature incubator for 14 days and then moved to MS solid medium with Cd concentrations of 0, 25, 50 and 75 uM. The plants were grown under a 16/8 h light and dark cycle and 23/20°C day/night temperature. After 10 days, we compared the growth status of wild-type (Col-1), deletion mutant (atha 4), and heterologous overexpression (*RcHA4*-OE) plants. We used the LEICA intelligent biological microscope (DM6) to observe and take pictures of the seedlings, and used LASX software to measure the lengths of plant roots, The average of 5 biological replicates was taken for each set of data. The deletion mutant atha4 (SALK_121317C) was purchased from the AraShare website (http://www.arashare.cn/)

### Data analysis

All the physiological index determination data are shown as the mean and standard deviation of three biological replicates. SPSS 19.0 software was used for independent sample T-tests. Graphpad software was used to draw bar charts, and significant differences are indicated.

## Results and discussion

In our first experiment, we compared the germination and growth of castor treated with water (as the control) or with three Cd solutions of 300 mg/L, 700 mg/L and 1,000 mg/L. We aimed to investigate the effect of Cd on the promotion or inhibition of growth, as well as Cd injury to castor plants. Fig 1 depicts the effect of increasing Cd concentrations on the growth of castor.

### Physiological study of castor plants responding to Cd stress

Physiological studies of castor's response to Cd stress have been documented in the scientific literature [31–33]. We repeated some of these experiments to verify whether the data obtained are consistent with expectations. However, we expanded on previous research by studying castor's overall response at the physiological, protein and metabolic level. This gives us greater insight into the molecular mechanisms of castor's tolerance to Cd on multiple levels.

We determined the Cd concentration in the roots, stems and leaves of CK, ZA, ZB and ZC experimental materials. As expected, with increasing Cd stress, the concentration of Cd detected in the roots, stems and leaves of castor also increased. However, the increase of Cd content in stems and leaves was not significant. In contrast, Cd significantly accumulated in the roots of castor beans, and the Cd content was 25.6 and 33.5 times that in stems and leaves, respectively (Fig 2). This indicates that under Cd stress, the root system, which first contacts pollutants in the soil, is the first plant organ to accumulate Cd, and it is also an effective barrier to Cd translocation to the above ground plant parts, which is consistent with the study of John et al [34]. Because Cd initially accumulates in the castor bean root, differential proteomics and comparative metabolomics analysis of castor bean roots is of greatest significance to understand the mechanism of castor bean's Cd stress response.

The effects of different concentrations of Cd stress on the physiological indicators of castor plants (Fig 3). Compared with the control, Malondialdehyde (MDA), Oxygen free radical (OFR) rate are positively correlated with the concentration of Cd up to a certain level. Thus, excessive Cd may cause damage to the cell by mediating excessive ROS production. The activities of superoxide dismutase (SOD) and catalase (CAT) in castor plant roots showed a gradual decreasing trend with the increase in Cd stress. We speculate that Cd stress affects the Redox status of castor plants. Under low levels of Cd stress, the high SOD and CAT enzyme activities

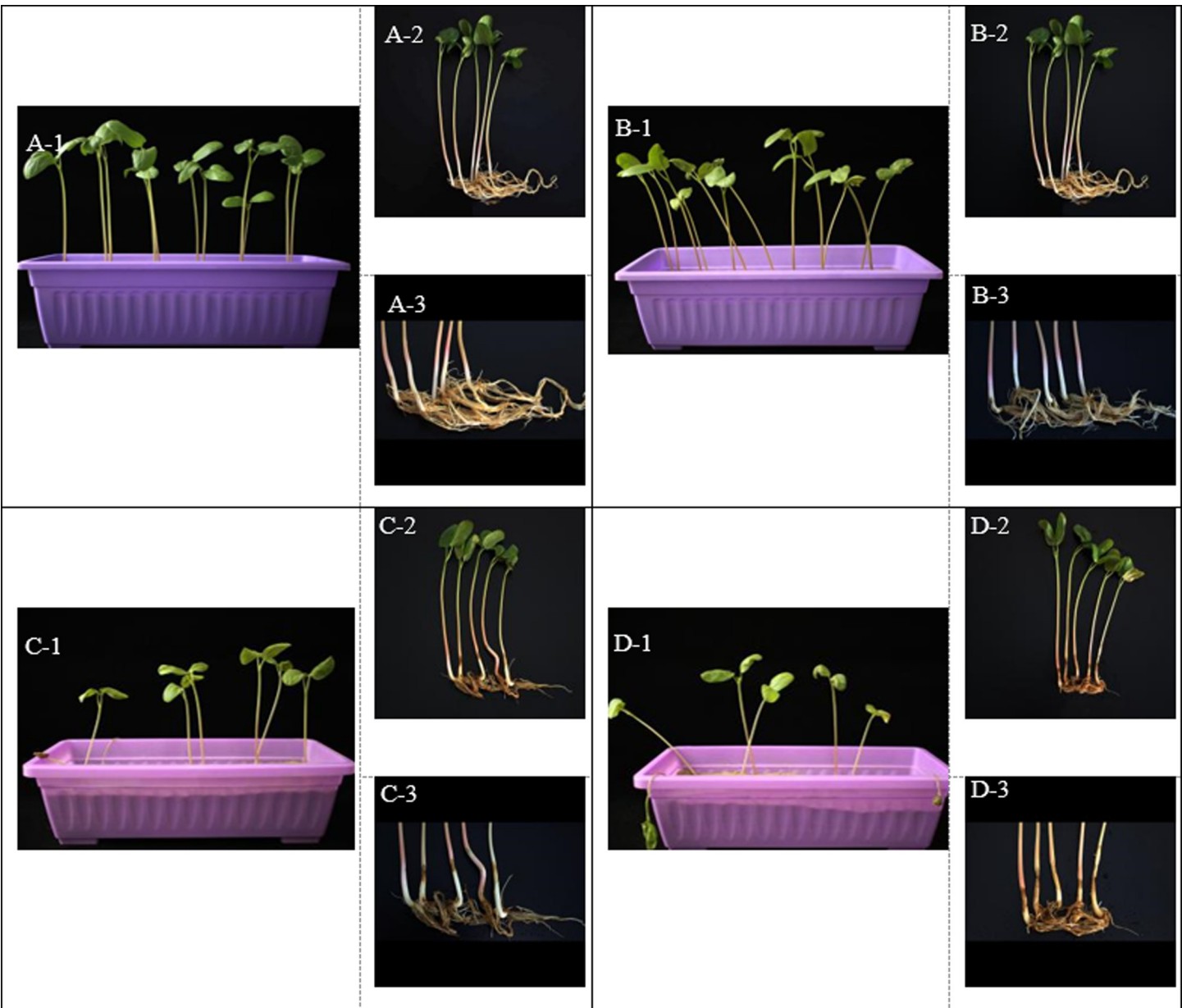

**Fig 1. Growth status of castor plant and root system under different doses of Cd stress.** Note: A1-A3: Intact plants and their roots under no Cd treatment; B1-B3: Intact plants and their roots under 300 mg/L Cd treatment; C1-C3: Intact plants and their roots under 700 mg/L Cd treatment Roots; D1-D3: Intact plants and roots under 1000 mg/L.

maintain the normal growth of castor plants through the decomposition and removal of ROS. With increasing Cd stress, the degree of damage of proteins involved in Redox is aggravated, which results in the decrease of the detoxification ability of Castor. In addition, with an increase in Cd stress, the activities of glutathione S-transferase (GST) and total antioxidant capacity (T-AOC) first increased and then decreased. For example, the highest GST and T-AOC activities were detected in the 300mg/L Cd-treated samples. GST, as a non-enzymatic antioxidant, forms a part of the antioxidant system with SOD, CAT and other antioxidant enzymes, and it plays an important role in promoting plant growth and improving its ability

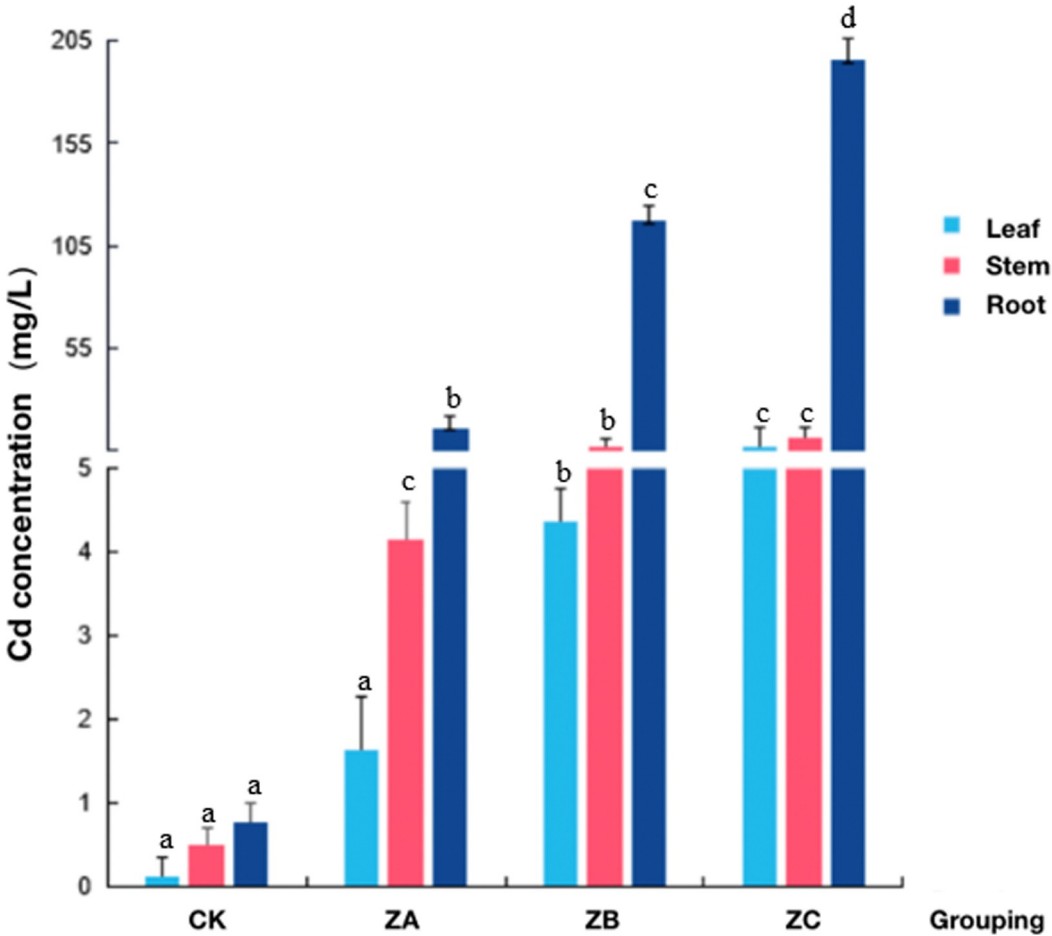

**Fig 2. Cd concentration determination results of castor plants treated with different heavy metal Cd concentrations.**

to tolerate stress [35]. The high activity of GST and T-AOC further confirms that castor can resist the damage caused by ROS by improving its own antioxidant capacity under the stress of low concentrations of Cd.

## Proteomics of Cd stress response in castor

In recent years, with the development of plant genome sequencing technology, a variety of other high-throughput omics technologies have also spread rapidly, and are widely used in the study of the molecular mechanisms of plants' response to heavy metal stress [36–38]. The use of these omics technologies can clarify the tolerance and detoxification mechanisms of plants in response to heavy metal stress from multiple perspectives Proteomics, especially, can analyze and determine the target proteins that actively respond to heavy metal stress [39].

In this study, based on the isotope-labeled relative quantification (iTRAQ) technology, a total of 217 and 156 protein spots were detected with significantly increased or decreased accumulation under different levels of Cd stress (The results are shown in S3–S5 Tables). All were classified into 7 functional categories (Fig 4): Stress/defense/detoxification (34%), Energy and carbon metabolism (21%), Protein folding and degradation (14%), DNA damage repair (4%), Amino acid metabolism (3%), Others (17%) and unknown function (6%).

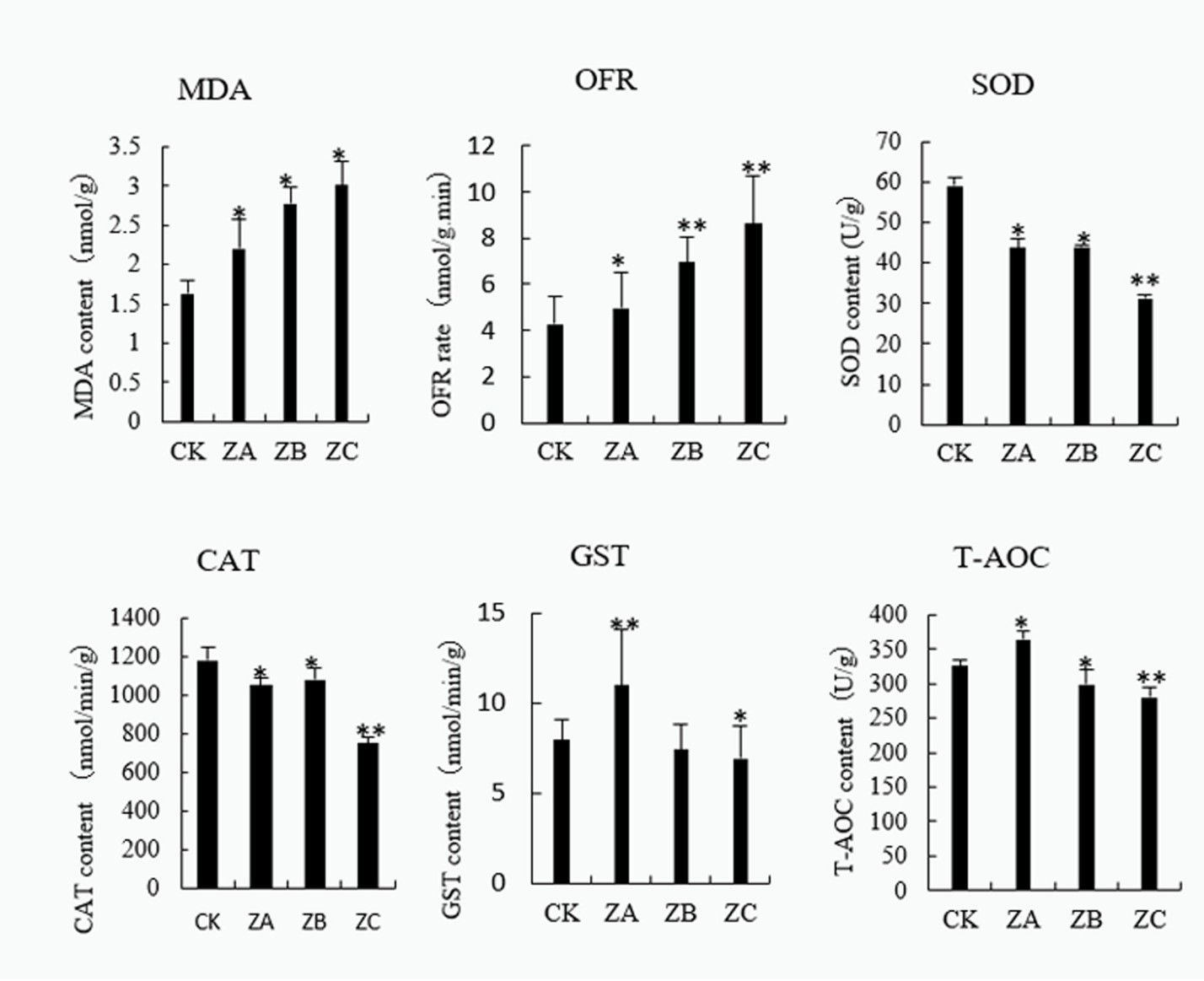

**Fig 3. Determination of physiological indicators.** Note: *: 0.01<P≤0.05, **: 0<P≤0.01.

A hierarchical cluster analysis of the differential proteins showed that the proteins accumulated differently under the stress of the three different concentrations of Cd (Fig 5). Here, the potential functions of the different proteins in each group were analyzed to explore the molecular mechanisms of castor's adaptation to, and defense against, the stress of different concentrations of Cd.

Under the stress of 300mg/L Cd, the main down-regulated proteins in castor roots include tubulin α chain, pectin esterase and other proteins involved in cytoskeleton synthesis. This result is consistent with another study [40]. During the process of uptake of Cd into cells, due to changes in ion homeostasis and other reasons, the structures of the plant cell wall and the cell membranes will be damaged. In addition, the amounts of ferredoxin -NADP⁺ reductase, citrate 1, 2-oxygenase and the accumulation of other proteins involved in photosynthesis and energy metabolism decreases, indicating that Cd stress caused changes in photosystems and

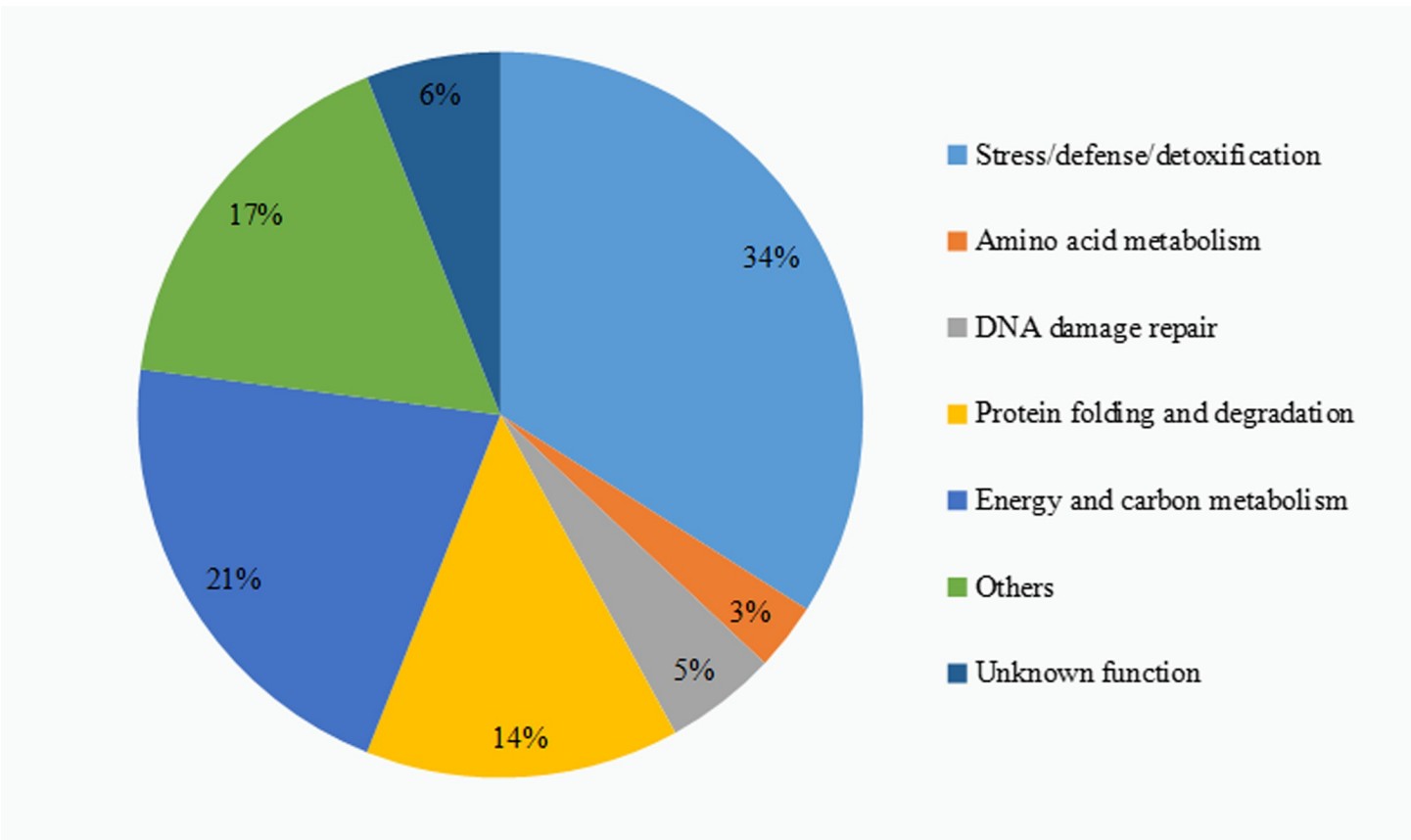

**Fig 4. Differential protein function classification in group Z_VS_C.**

inhibited related processes of energy metabolism. The proteins whose content increased mainly include: 40S ribosome binding protein S2, 60S ribosome binding protein L3, heat shock protein and other proteins related to damage repair. The increased amounts of these proteins suggests that castor is involved in cellular damage repair by improving ribosome translation efficiency and ensuring correct protein folding. Epoxy hydrolase, succinate dehydrogenase, cytochrome b5 protein (Cyt-b5), cytochrome C oxidase and other proteins involved in the tricarboxylic acid cycle and cellular oxidative phosphorylation and other pathways are also significantly up-regulated. On the one hand, these proteins participate in various redox reactions in the cell to regulate the balance of ROS in castor plants. On the other hand, they can participate in the cellular oxidative phosphorylation pathway and synthesize ATP for direct use by plants. The increase in the activity of these proteins likely plays an important role in resisting Cd stress. It is worth noting that the expression of proteins involved in lignin synthesis such as O-methyltransferase and epimerase are specifically up-regulated. Studies have found [41–43] that lignin is an important component of plant secondary cell walls. Not only can it protect the normal functions of plants, but it can also enhance the plant's resistance to stress by enhancing the water barrier and mechanical strength of plant cell walls.

Under 700 mg/L of Cd stress, the number of down-regulated proteins involved in cell structure synthesis, Redox, energy metabolism, DNA damage repair and protein misfolding prevention in castor root increased significantly. In addition, we found that the expression of the Sco1 protein and other proteins related to the cell's inner membrane were down-regulated. This means that high concentration of Cd stress seriously damages castor cell structure and

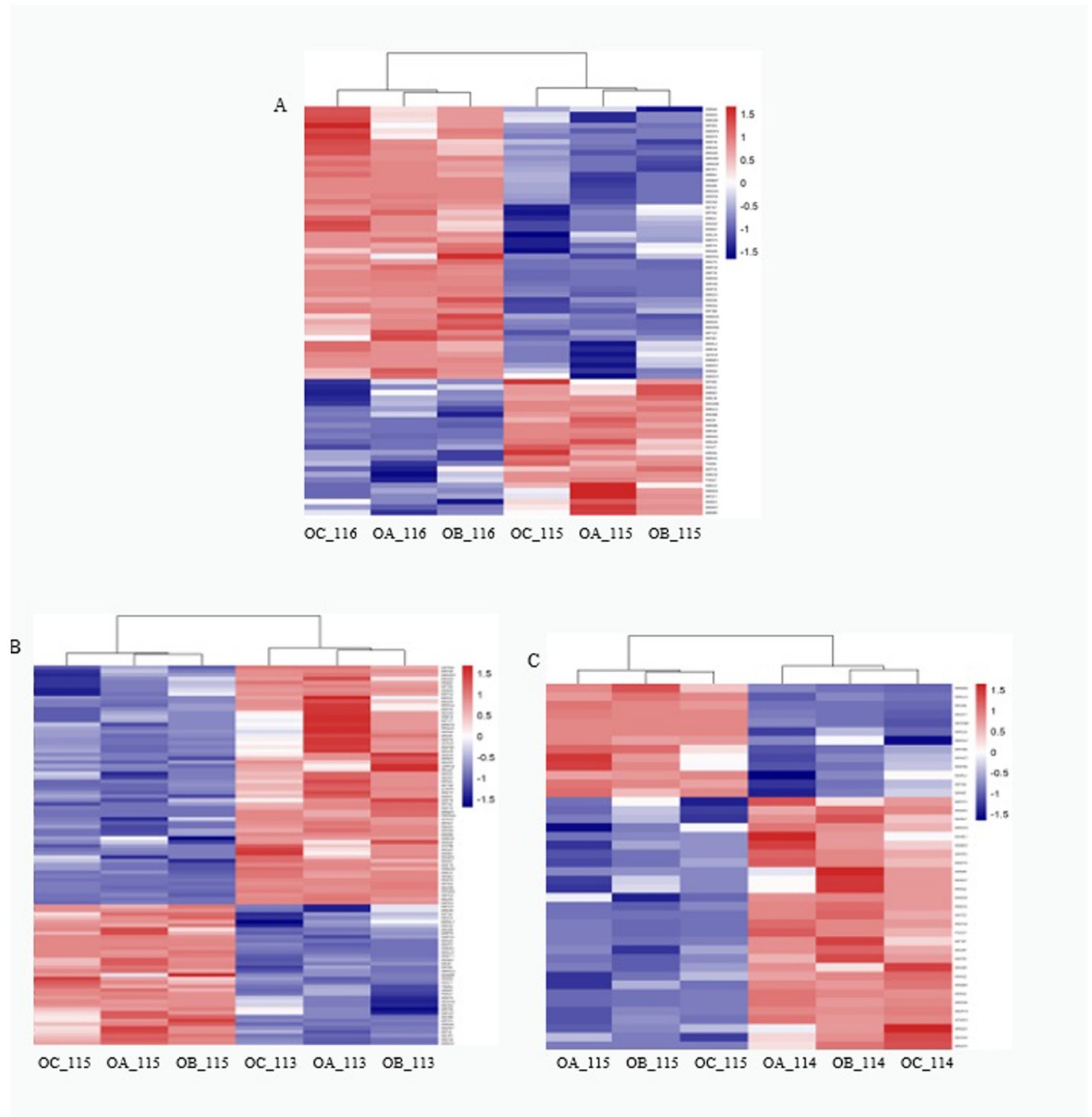

**Fig 5. Differential protein cluster analysis.** Note: A: ZA_VS_CK differential protein cluster analysis; B: ZB_VS_CK differential protein cluster analysis; C: ZC_VS_CK differential protein cluster analysis.

inhibits its redox energy metabolism and other reaction processes, leading to excessive accumulation of intracellular ROS and damage to the membrane structures of various organelles. Analysis of up-regulated proteins found that the accumulation of proteins related to soy

protein and gluten, that are involved in nutrient storage, was increased. We speculate that castor may enhance its tolerance to Cd by improving nutrient accumulation. 2-Cys peroxidase, superoxide dismutation enzymes are important antioxidant enzyme systems in plants, and the accumulation of these proteins plays an important role in eliminating ROS in cells. This result is also consistent with the physiological results. In contrast to the ZA group, the proteins involved in plant photosynthesis were found to be significantly up-regulated in castor under 700 mg/L Cd stress, including (S)-2-hydroxy acid oxidase, 1,5-diphosphate ribulose carboxylase/oxygenase (Rubisco) and other key enzymes involved in the carbon fixation process. Related enzymes involved in chlorophyll synthesis such as 4-hydroxy-3-methylbut-2-enyl diphosphate reductase (HDR) and 2-alkenyl diphosphate reductase [44] also increased. It is worth noting that the expression of $K^+$ channel protein [45], spermidine synthase1[46], ammonium transporter [47] and other proteins related to the regulation of cellular ion homeostasis and osmosis were also up-regulated in ZB. It is known that roots exposed to Cd stress will accumule potassium ions. At the same time, $Cd^{2+}$ inhibits the absorption of $Ca^{2+}$, changes the homeostasis of $Ca^{2+}$ and destroys the cells' signal transduction pathway [48]. In this case, the upregulation of the $K^+$ channel protein's expression level plays an important role in promoting $K^+$ absorption, plant growth and photosynthetic rate [49]. A study found [50] that exogenous polyamine (PA) as a nitrogenous base to promote plant growth and development, plays an important role in improving plant resistance to adversity stress. Spermine synthase is one of the key enzymes in polyamine metabolism [51]; Ammonium transporter [52], as an important material transport protein in plants, promotes plant growth and development by participating in the effective absorption of $NH4^+$ and other inorganic nitrogen to cope with a variety of abiotic stressors [53]. It may play a role in cadmium resistance.

Under Cd stress of 1,000 mg/L, consistent with the above results, related proteins involved in cell structure synthesis, oxidative system and damage repair are the main down-regulated proteins. In addition, it was observed that β -hexosaminase and other proteins involved in the hydrolysis of polysaccharides were significantly decreased. Since polysaccharides are widely present in the cytoskeleton and cell wall of plants, castor plants may improve their resistance to Cd stress by increasing the mechanical strength of the cell wall. Among the up-regulated proteins, related proteins involved in lignin synthesis, antioxidant enzyme system, photorespiration pathway and ion homeostasis showed a certain amount of accumulation, which is consistent with the results of the other two treatment groups. However, under 1,000 mg/L Cd stress, UDP-glucose 4-epimerase that catalyzes galactose metabolism,CASP-like protein 1D1 [54], that is involved in programmed cell death, protein [55], that is induced by acute trauma, and alcohol dehydrogenase, that is induced [56,57] under hypoxic conditions, were all significantly up-regulated. Among these, UDP-glucose 4-epimerase catalyzes galactose metabolism to generate ATP, while the generated metabolite UDP-glucose could further participate in the metabolic pathways such as glucuronic acid and pentose phosphate and participate in plant stress. The expression level of alcohol dehydrogenase that is involved in the short-chain alcohol metabolism pathway was significantly up-regulated. We speculate that excessive Cd induced production of a large number of ROS, thus, severely damaging the photoreaction system in photosynthesis, and leading to a large amount of $CO_2$ accumulation in cells. Alcohol dehydrogenase, a key enzyme in the anaerobic metabolic pathway, may be involved in $CO_2$ fixation in some way to participate in the castor bean response to Cd stress. The increase in the amounts of oxidative stress proteins such as CASP-like protein 1D1 and wound-induced protein may be due to the fact that when castor oil plants are more severely damaged, they respond to Cd stress by activating their immune defense system and activating the programmed cell death process in necrotic cells.

**RT-qPCR analysis of candidate Cd-stress genes.** On the basis of the above analysis, 9 proteins with significant expression differences and possibly participating in the response to Cd stress were selected for RT-qPCR analysis, The primer sequence is shown in Appendix S4 Table. The results confirmed that the transcriptional differences in the expression of the genes encoding the Cd stress response proteins is consistent with our proteomics results (Fig 6). That is to say, we can study the response mechanisms of castor plant roots to Cd stress at the transcriptional level starting from the different protein coding genes.

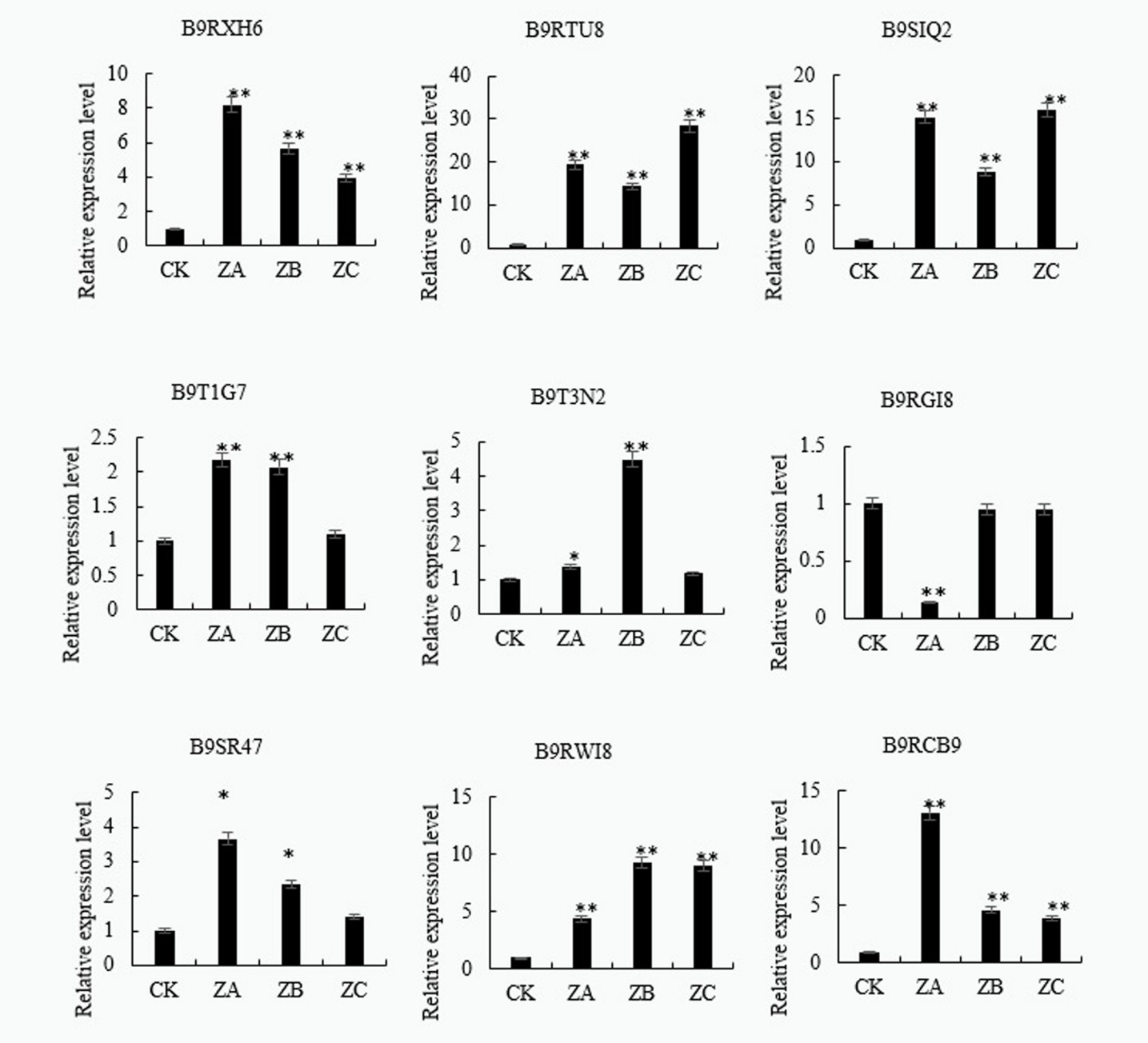

**Fig 6. RT-qPCR detection results of some differential proteins in castor plants in response to Cd stress.** Note: *: $0.01 < P \leq 0.05$, **: $0 < P \leq 0.01$.

**Metabolite analysis.**    As an emerging research technology, metabolomics can systematically explore the distribution of metabolites in organisms, and at the same time conduct qualitative and quantitative analysis of metabolites involved in major metabolic pathways such as sugars, alcohols, amino acids, organic acids, and others. To understand biological growth and development, stress response and other molecular mechanisms via metabolomics is of great value and significance [58–60]. Combined with metabonomics technology to study the response of castor plants to different levels of Cd stress, it is helpful to understand the correlation between Cd stress and the metabolic pathways of castor.

In order to understand the overall metabolic difference between the samples, principal component analysis was performed on the samples of each group (Fig 7). Control group CK (CK1, CK2, CK3) and treatment group ZA (ZA1, ZA2, ZA3), ZB (ZB1, ZB2, ZB3), ZC (ZC1, ZC2, ZC3) are distributed in the PCA chart, indicating that differences among the control group and the treatment groups were detected at the metabolite level. In addition, the separation trend of samples in each group was significant and the sample repetition between the groups was good, indicating that the metabolites between the groups changed significantly, and the data obtained are reliable.

Based on the UPLC-MS/MS detection platform and a self-built database, a total of 72 different metabolites (The resultare shown in S6–S8 Tables) were screened. The metabolites whose abundances were differentially affected by the three Cd stress treatments and their overlapping Venn diagrams are shown in Fig 8. Under 300mg/L Cd stress treatment, 15 and 25 common metabolites increase and decrease in abundance, respectively. With increasing Cd stress, the number of common metabolites that reduce in abundance gradually increases. Especially under the highest Cd stress of 1,000 mg/L, the largest number of metabolites decrease in abundance.

The types and amounts of metabolites that are affected by different levels of increasing Cd stress in castor plant roots were analyzed (Fig 9). In general, metabolites such as lipids, organic acids and flavonoids were more affected by Cd stress. Comparing the metabolite amounts found in response to the different Cd stresses to the control (ZA_VS_CK, ZB_VS_CK and ZC_VS_CK), we found that the levels of lipids, amino acids and their derivatives and organic

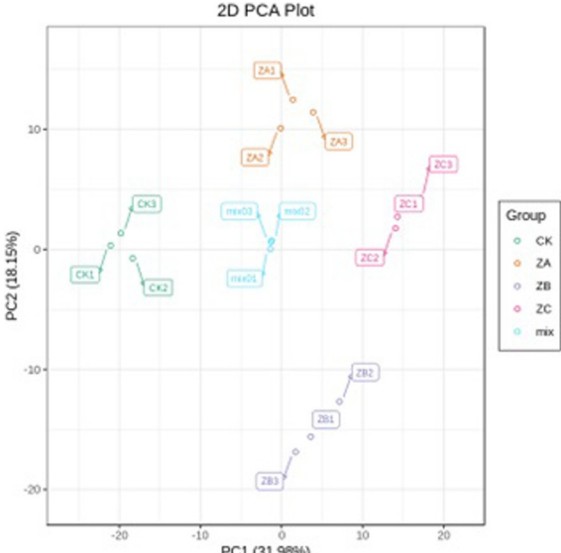

**Fig 7. PCA score chart of mass spectrum data of each group of samples.** Note: X axis represents the first principal component, Y axis represents the second principal component.

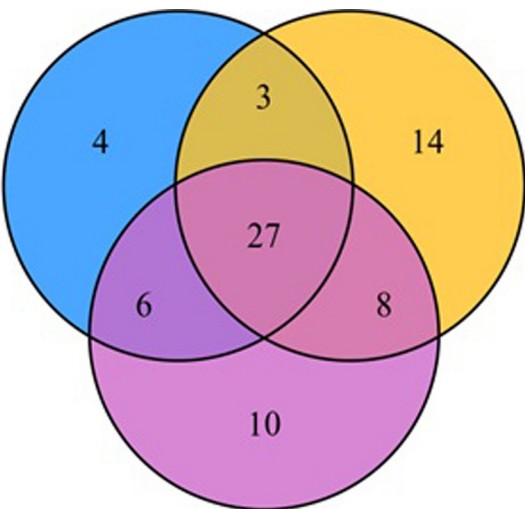

**Fig 8. Venn diagram with different components.**

acid metabolites were all reduced, and the degree of reduction increased as the Cd stress increased. This is different from the results of "low promotion and high inhibition" of organic acids, amino acids and other metabolites in response to Cd stress proposed in studies on rice [61] and Eichhornia crassipes [62]. On the one hand, different plants may respond to Cd stress in different ways. On the other hand, the differences may be because the Cd stress concentrations used in this study were significantly higher than in the other studies. In contrast to the above, the levels of polyphenol metabolites involved in lignin synthesis, such as tannins and phenolic acids, were significantly increased [63–65] upon increasing Cd stress in castor roots. This was highly consistent with the results of our proteomics study, Namely, we believe the increase in lignin precursor metabolite levels indicates that castor enhances its tolerance to Cd stress by blocking the absorption of $Cd^{2+}$ by synthesizing lignin and improving the strength and toughness of its secondary cell walls. It is worth noting that, compared with the significant increase in the ZA and ZB groups, the flavonoid metabolite levels in the ZC group showed a certain downward trend. Another study found [66] that flavonoid metabolites can form a stable chelate with $Cd^{2+}$. This chelate prevents the excessive accumulation of ROS in cells from causing oxidative damage to plants by inhibiting the generation of oxygen free radicals catalyzed by $Cd^{2+}$. Therefore, the significant increase in flavonoids in the ZA and ZB groups indicates that castor plants maintain normal growth by inhibiting and eliminating intracellular ROS under the stress of 300 mg/L and 700 mg/L Cd, and this result also appeared at the physiological and protein level. The downregulation of flavonoids in the ZC group may be caused by excessive damage to castor plants caused by the very high Cd concentration of 1,000mg/L.

In addition, combined with KEGG functional annotation and enrichment results of metabolites with significant differences among all groups, we found that under the three different doses of Cd stress, the significant differences in the roots of castor plants were all enriched in the categories of ABC transporters, Biosynthesis of amino acids, Aminoacyl–tRNA biosynthesis, Carbon metabolism, 2–Oxocarboxylic acid metabolism and other pathways. The above results may indicate that castor responds to Cd stress mainly by promoting protein biosynthesis, energy metabolism, photorespiration pathway, nitrogen synthesis, biotin accumulation and transport [67,68]. This is highly consistent with our proteomics results. Combining the different metabolic pathways involved in each group found that under 700mg/L Cd stress, taurine produced by the taurine pathway was specifically increased in castor roots. Taurine may

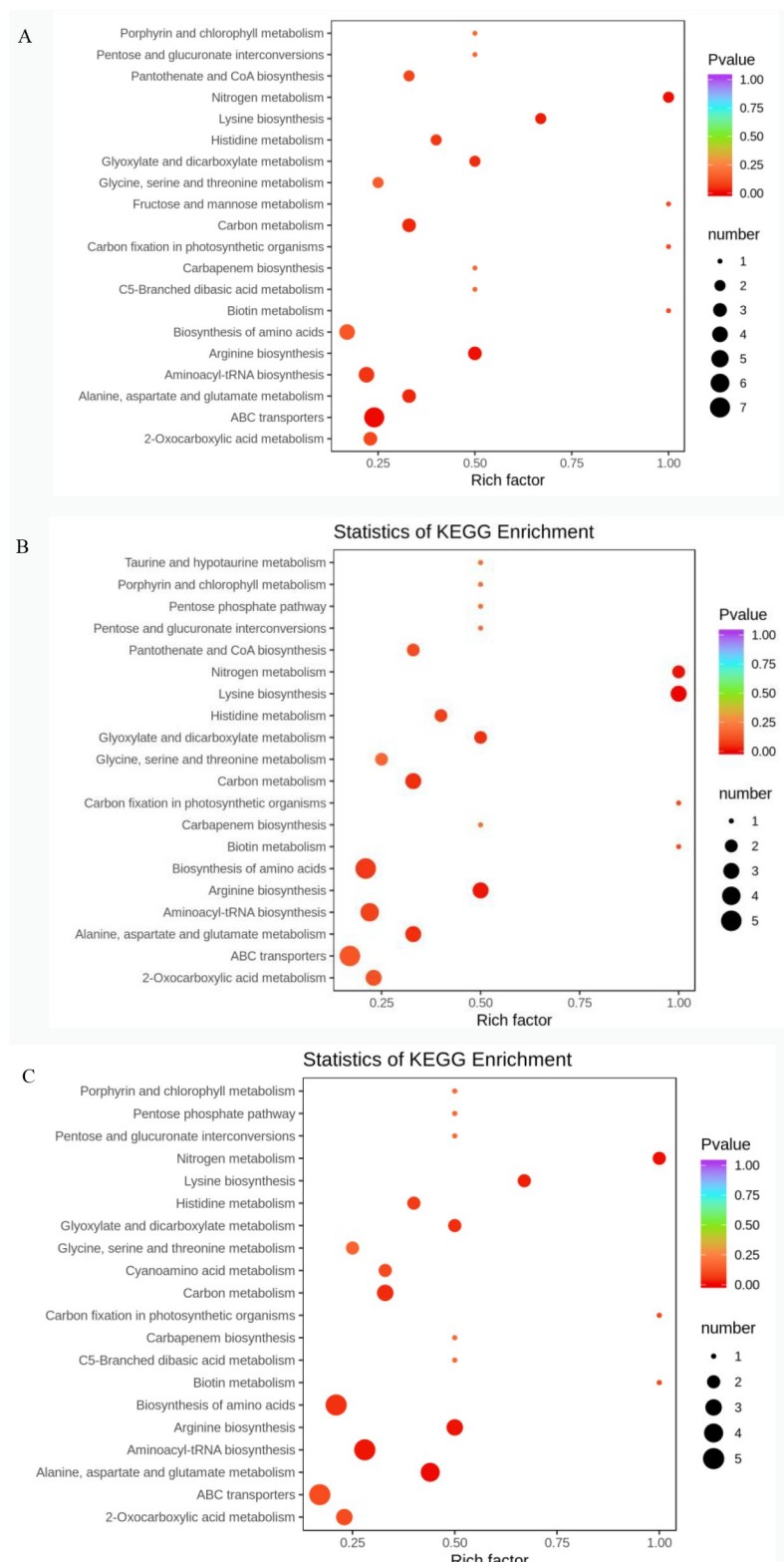

**Fig 9. Enrichment analysis of KEGG metabolic pathway of differential metabolites in Z_VS_ C group.** Note: A: KEGG differential enrichment bubble chart in ZA_VS_CK; B: KEGG differential enrichment bubble chart in ZB_VS_CK; C: KEGG differential enrichment bubble chart in ZC_VS_CK.

play an important role in improving castor plants' response to the higher concentration of Cd stress (700mg/L) by participating in the detoxification of Cd [69]. Moreover, the metabolic pathways of pentose and glucuronic ester transformations in castor roots under 1,000mg/L Cd stress also verified the specific accumulation of UDP-glucose 4-epimerase in our proteomics study.

## Combined analysis of differential proteomics and comparative metabolomics

In order to further determine the effects of different levels of Cd stress on the metabolites in the roots of castor and the response of the castor metabolic pathways to Cd stress, the differential proteomics analyses of ZA_VS_CK, ZB_VS_CK, and ZC_VS_CK were combined with the comparative metabolomics analyses. We used the KEGG database to analyze the metabolic pathways in which the proteins and metabolites respond together to Cd stress.

The combined analysis of the different sets of proteomics data and comparative metabolomics data found that a total of 7 different proteins and 7 different metabolites in ZA_VS_CK were annotated in the same metabolic pathway, including Pantothenate and CoA. biosynthesis, Aminoacyl-tRNA biosynthesis, and the Pentose phosphate pathway (Fig 10A). A total of 8 differential proteins and 9 differential metabolites in ZB_VS_CK were annotated in the same metabolic pathway, including Cysteine and methionine metabolism, Carbon fixation in photosynthetic organisms, Carbon metabolism, Arginine and proline metabolism, Glutathione metabolism, Glyoxylate and dicarboxylate metabolism. Among these, metabolites and proteins are highly enriched in carbon metabolism, glyoxylic acid and dicarboxylic acid ester metabolic pathways (Fig 10B). A total of 14 differential proteins and 4 differential metabolites in ZC_VS_CK were annotated in the same metabolic pathway, including Biosynthesis of secondary metabolites, Flavonoid biosynthesis, Glyoxylate and dicarboxylate metabolism, and Carbon metabolism (Fig 10C).

Combined with our analysis of the proteomic and metabolomic results under 300 mg/L Cd stress, we conclude that castor plants mainly maintain normal growth by promoting energy metabolism, protein synthesis, and repair. With increasing Cd concentration, the damage to castor plants was aggravated. While maintaining the energy required by the organism, castor plants alleviated the toxic damage of $Cd^{2+}$ by increasing the photosynthetic rate and the synthesis of detoxification substances such as proline, arginine and glutathione. When the Cd stress increased to 1,000 mg/L, castor repaired the oxidative damage to itself by increasing the production of secondary metabolites such as flavonoids.

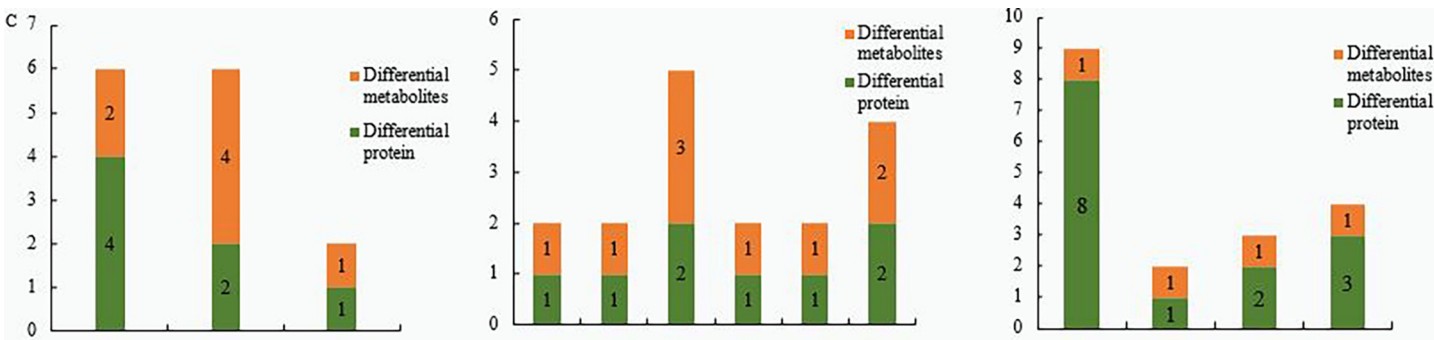

**Fig 10. Metabolic pathways involving differential proteins and metabolites in Z_VS_ C group.** Note: A: Metabolic pathways involving differential proteins and metabolites in ZA_VS_ CK; B: Metabolic pathways involving differential proteins and metabolites in ZB_VS_ CK; C: Metabolic pathways involving differential proteins and metabolites in ZC_VS_ CK.

## Subcellular localization

In order to determine the cell specificity of *RcHA4* gene expression, we constructed the subcellular localization transient expression vector pBI121-*RcHA4*-GFP to produce an N-terminal green fluorescent protein fusion with *RcHA4*. We transiently expressed this construct in onion epidermal cells and observed the results under a fluorescence microscope. The results showed that under green fluorescence, the pBI121-*RcHA4*-GFP fusion protein accumulated significantly in the cell membrane (Fig 11).

Homozygous *Arabidopsis* positive plants were identified at the DNA level, and the positive overexpression plants *RcHA4*-OE were isolated (Sequencing results are shown in Appendix S9 Table). RT-qPCR analysis of the positive plants found (Fig 12) that the gene *RcHA4* was significantly induced under 50 μM Cd stress, which was about 2.9 times that of the control. This provides suggestive evidence that *RcHA4* plays an important role in the response mechanism of castor plants to Cd stress. However, when the concentration of Cd increased to 500 μM, the relative expression level of *RcHA 4* gene was not significantly up-regulated. This may be because this degree of Cd stress leads to irreversible damage, which the plants cannot survive.

## Cd tolerance analysis

We used a 10X intelligent biological microscope to observe the growth of Col-1, *Atha4*, and *RcHA4*-OE, and used LASX software to measure the root length and lateral root number of each *Arabidopsis* plant (Table 1 and Fig 13). The results showed that under Cd-free treatment, Col-1, *Atha4* and *RcHA4*-OE grew equally well, with well-developed root systems and lush leaves without obvious damage. Under 25 μM Cd stress, the relative lengths of the roots of Col-1 and *Atha4* were significantly shorter than that of *RcHA4*-OE. Among these lines, the leaf growth of *Atha4* was inhibited and the leaves of individual plants showed obvious chlorosis and wilting. Under 50 μM Cd treatment, the root lengths and the number of lateral roots of

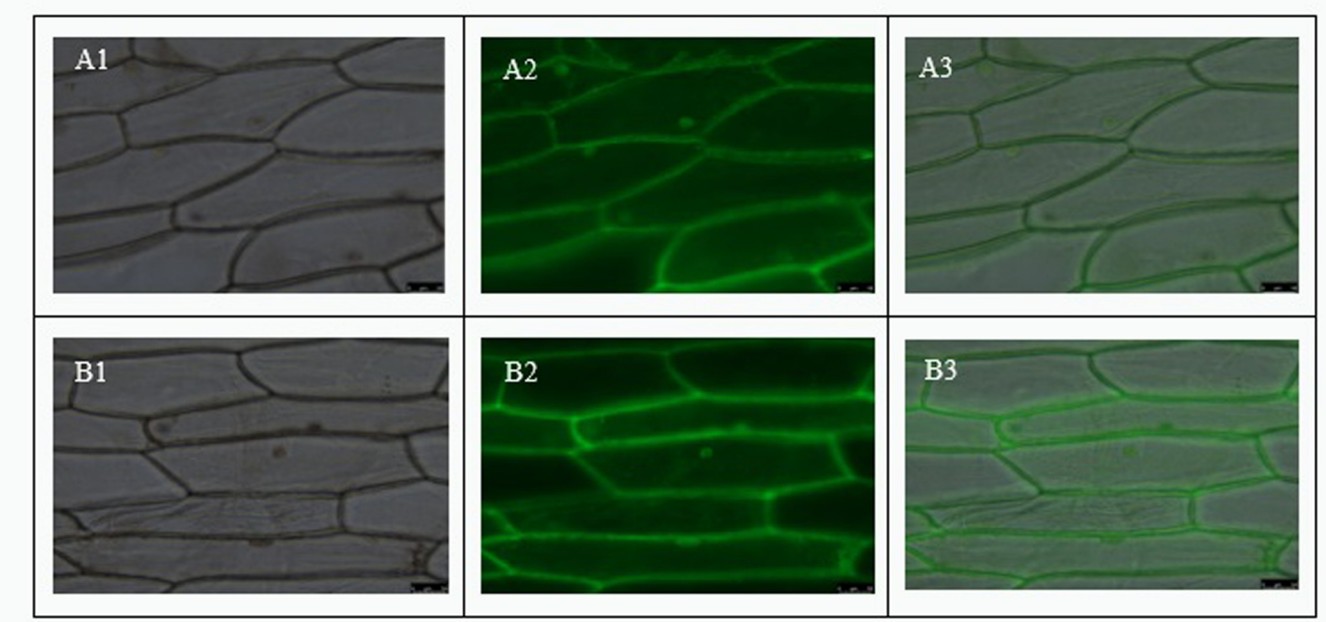

**Fig 11. Observation of subcellular localization.** Note: A1, A2, A3: Protein expression results of non-transformed cells under bright field, green fluorescence, and superimposed conditions; B1, B2, B3: Expression results of RcHA 4 protein under bright field, green fluorescence, and superimposed conditions.

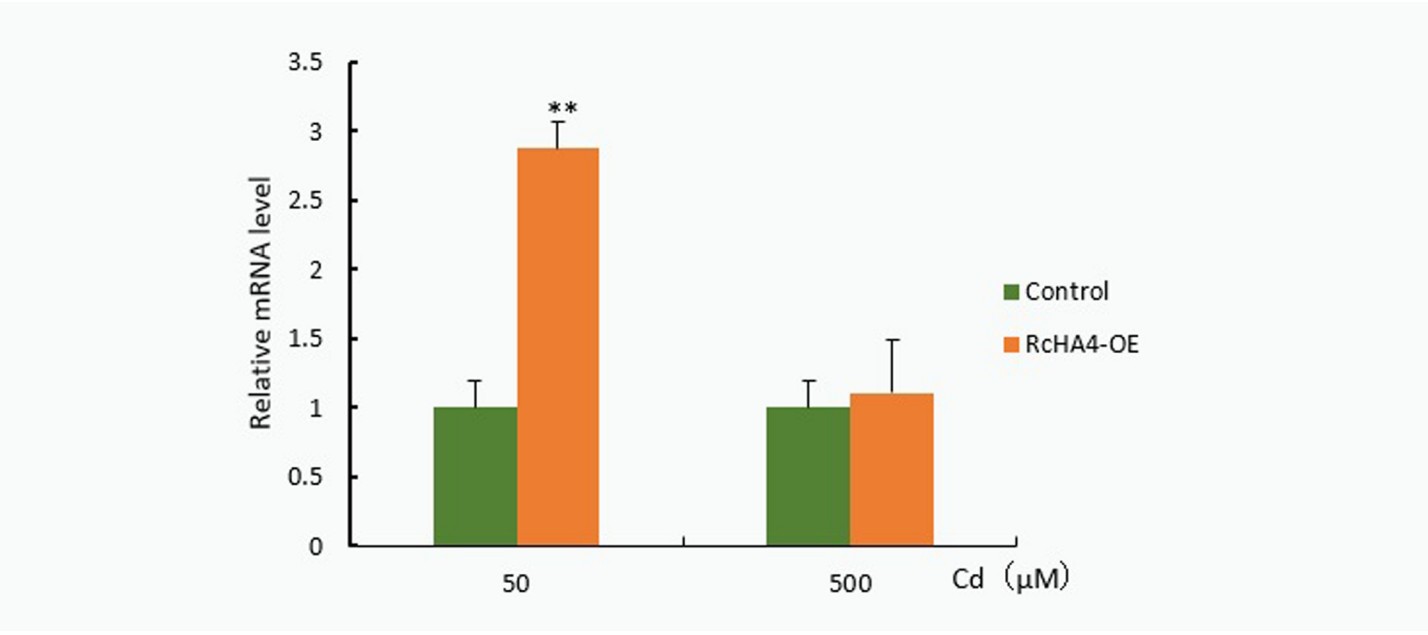

**Fig 12. qRT-PCR analysis showed the expression of *RcHA4*.** Note: *: 0.01<P≤0.05, **: 0<P≤0.01.

*RcHA4*-OE were significantly better than those of Col-1 and *Atha4*. At this Cd concentration, the degree and area of yellowing on the leaves of *Atha4* increased, and the leaves of Col-1 began to appear chlorotic. When the Cd stress concentration was increased to 75 μM, Col-1, *Atha4*, and *RcHA4*-OE were all seriously damaged. By comparing the length of the roots, the number of leaves and the degree of damage, *RcHA4*-OE still appeared more resistant to Cd than Col-1, and *Atha4*. Under Cd stress, the relative expression of *RcHA4* in castor is up-regulated, indicating that castor plants enhance their tolerance to Cd stress by increasing the expression of *RcHA4*.

At present, the molecular mechanisms of phytochelatins (PCs), metallothionein (MTs), ATP-binding cassette (ABC) transporters, heavy metal transporting ATPase (HMAs) transporters and other proteins in response to heavy metal Cd stress through chelation have been

**Table 1. The statistical results of the root length and lateral root number of Col-1, atha 4, *RcHA 4*-OE under different Cd stress concentrations.**

| Cd concentration (μM) | Mate-rial | Linoleic Acid (cm) | Linolenic Acid (piece) |
|---|---|---|---|
| 0 | Col-1 | 32.24±3.31a | 22±4a |
| | atha 4 | 32.78±4.25a | 20±3b |
| | *RcHA 4*-OE | 24.27±4.81b | 20±4c |
| 25 | Col-1 | 21.51±3.46c | 15±2a |
| 25 | atha 4 | 18.39±2.36b | 13±4b |
| | *RcHA 4*-OE | 21.29±3.63a | 17±3c |
| 50 | Col-1 | 20.73±3.74a | 12±3a |
| | atha 4 | 15.46±4.13b | 13±3b |
| 50 | *RcHA 4*-OE | 19.43±3.21c | 15±4c |
| 75 | Col-1 | 18.01±3.43b | 01±1a |
| | atha 4 | 14.28±2.14c | 02±2b |
| | *RcHA 4*-OE | 20.73±1.72a | 02±2b |

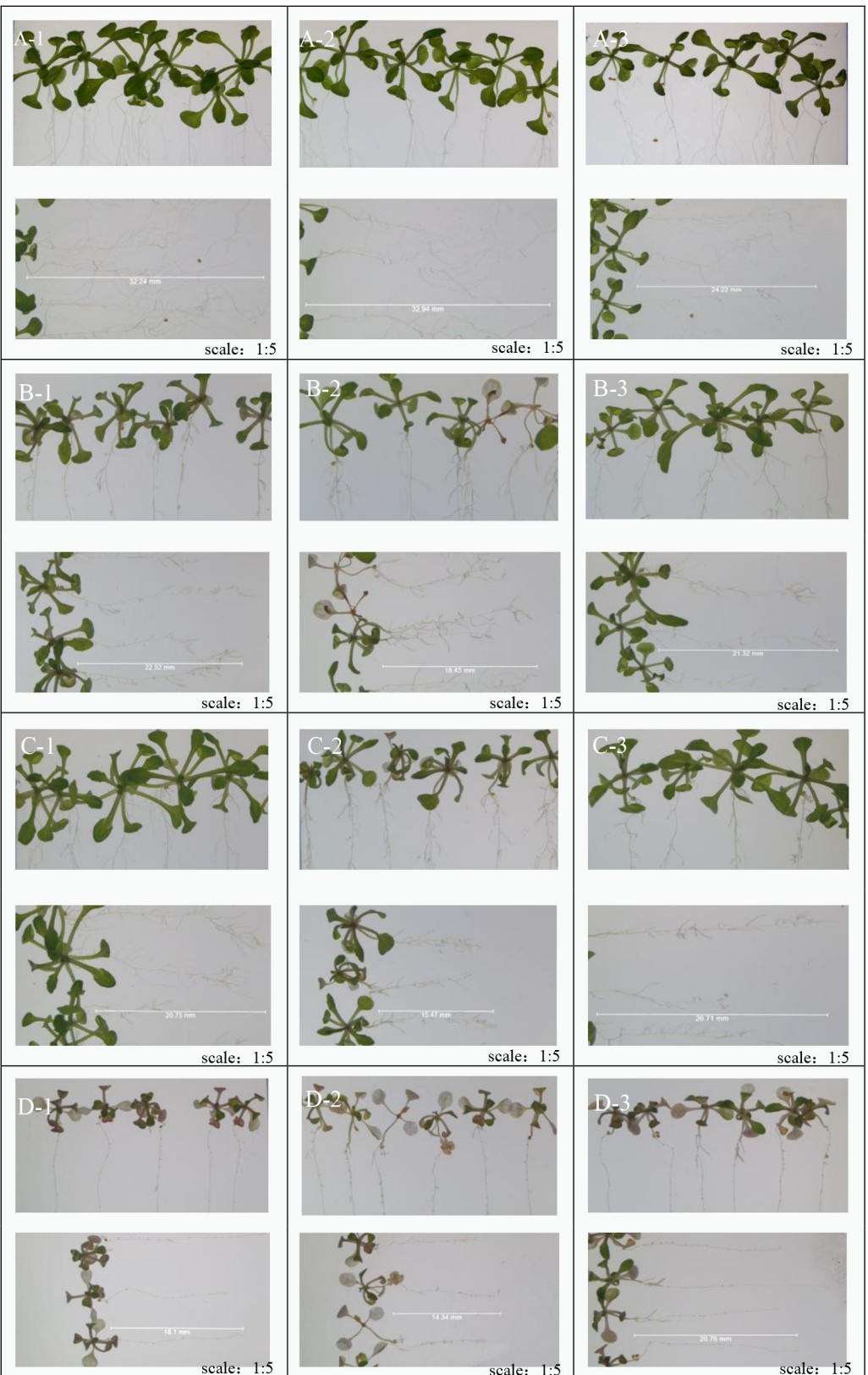

**Fig 13. Analysis results of tolerance of Col-1, atha 4, *RcHA 4*-OE to heavy metal Cd.** Note: A1-A3: Col-1, atha 4, *RcHA 4*-OE under no Cd stress; B1-B3: Col-1, atha 4, *RcHA 4*-OEunder 25 μM Cd stress; C1-C3: Col-1, atha 4, *RcHA 4*-OE, under 50 μM Cd stress *RcHA 4*-OE; D1-D3: Col-1, atha 4, *RcHA 4*-OE under 75 μM Cd stress.

confirmed by most studies [70], but the molecular mechanism related to the plasma membrane ATPase's response to heavy metal Cd stress is still not clearly understood. In this study, through proteomics and RT-qPCR verification, a plasma membrane ATPase was selected for functional verification of its role in Cd stress tolerance. *RcHA4* was chosen for detailed study due to its high expression at both the transcriptional and protein levels, and it was hoped to find and explore castor's specific molecular mechanisms involved in the detoxification of Cd.

As a cell membrane protein, plasma membrane ATPases play important roles in plant growth and development as well as in responses to stresses. Studies [71–73] found that plasma membrane ATPase catalyzes ATP hydrolysis on one side of the plasma membrane, and the energy generated is involved in the movement of $H^+$ in the cytoplasm to the outside of the membrane to form $H^+$ gradients while maintaining plant growth. Excessive secretion of $H^+$ will cause excessive acidification of the cell wall, thus promoting elongation and growth of cells. When plasma membrane ATPase hydrolysis occurs in guard cells, the hydrolyzed ΔPH can induce changes in guard cell turgor, which adjusts the opening and closing of stomata. In addition, Ohno T proposed [74] that the plasma membrane ATPase is involved in regulating the secretion of organic acids and other detoxification metabolites in plants. Our study found that $Cd^{2+}$ enters the cell through the castor oil plant root system, which seriously damages the ion homeostasis of the castor plant. Increasing the duration of the stress and the level of the Cd pollutant leads to excessive accumulation of ROS in cells, which further affects the plant's anti-oxidant system and photosynthesis. Castor mainly enhances its Cd tolerance by improving its energy metabolism, enhancing the photosynthetic carbon cycle and increasing the secretion of a variety of detoxification metabolites. Based on the above results, we speculate that plasma membrane ATPase *RcHA4* may be involved in the mechanism of castor's response to Cd stress by maintaining ion homeostasis, increasing energy metabolism, promoting photosynthesis, etc. Future studies will need to examine this in more detail for further confirmation.

## Conclusion

In combination with the results of this study, the root system of castor, as the main tissue that encounters Cd in the environment, and transfers Cd to the stem and leaf system. Physiological, proteomic, and metabolomic analyses emphasized that exposure to Cd stress resulted in changes in the Redox system of castor plants, dysregulation of ATP synthesis, and disruption of ion homeostasis. This is the main cause of growth inhibition of castor plants under Cd stress. Meanwhile, We identified and expressed a gene in *Arabidopsis* that significantly enhances plant Cd tolerance (*RcHA4*). This gene may be used to breed castor plant varieties with high Cd tolerance. This study has deepened the understanding of the mechanism of castor plants' response to Cd stress, and provided a valuable theoretical basis for the genetic identification of castor plants with superior soil heavy metal remediation abilities.

## Supporting information

**S1 Table. RT-qPCR primer sequence.**
(DOCX)

**S2 Table. Primer sequences of PCR and RT-qPCR.**
(DOCX)

**S3 Table. Identification results of differential proteins in the roots of ZA_VS_CK castor plants.**
(DOCX)

**S4 Table. Identification results of differential proteins in the roots of ZB_VS_CK castor plants.**
(DOCX)

**S5 Table. Identification results of differential proteins in the roots of ZC_VS_CK castor plants.**
(DOCX)

**S6 Table. Identification results of differential metabolites in the roots of ZA_VS_CK castor plants.**
(DOCX)

**S7 Table. Identification results of differential metabolites in the roots of ZB_VS_CK castor plants.**
(DOCX)

**S8 Table. Identification results of differential metabolites in the roots of ZC_VS_CK castor plants.**
(DOCX)

**S9 Table. Gene sequence alignment results.**
(DOCX)

## Acknowledgments

The authors thank AiMi Academic Services (www.aimieditor.com) for the English language editing and review services. Meanwhile, we would like to thank Qingdao Kechuang Biological Company for the analysis of proteomics and metabolomics in this study.

## Author Contributions

**Conceptualization:** Zhao Huibo, Huang Fenglan.

**Data curation:** Zhao Huibo, Zhao Yong, Luo Rui, Li Guorui, Wen Qi, Liang Xiaotian, Wang Zhiyan.

**Formal analysis:** Zhao Huibo, Zhao Yong, Liang Xiaotian, Yin Mingda, Wen Yanpeng.

**Methodology:** Zhao Huibo, Zhao Yong, Luo Rui, Li Guorui, Di Jianjun.

**Project administration:** Huang Fenglan.

**Writing – original draft:** Zhao Huibo, Zhao Yong, Luo Rui, Li Guorui, Di Jianjun, Wen Qi, Liang Xiaotian, Yin Mingda, Wen Yanpeng, Wang Zhiyan.

**Writing – review & editing:** Zhao Huibo, Zhao Yong, Huang Fenglan.

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
