## [Decision Letter · Decision Letter 0]

26 Sep 2022

PONE-D-22-20884Analysis of the mechanism of castor tolerance to Cd metal based on proteomics and metabolomicsPLOS ONE

Dear Dr. Fenglan,

Thank you for submitting your manuscript to PLOS ONE. After careful consideration, we feel that it has merit but does not fully meet PLOS ONE’s publication criteria as it currently stands. Therefore, we invite you to submit a revised version of the manuscript that addresses the points raised during the review process.

We look forward to receiving your revised manuscript.

Kind regards,

Ying Ma, Ph.D.

Academic Editor

PLOS ONE

Journal Requirements:

Reviewers' comments:

Reviewer's Responses to Questions

**Comments to the Author**

1. Is the manuscript technically sound, and do the data support the conclusions?

Reviewer #1: Partly

Reviewer #2: Yes

2. Has the statistical analysis been performed appropriately and rigorously? 

Reviewer #1: Yes

Reviewer #2: Yes

3. Have the authors made all data underlying the findings in their manuscript fully available?

Reviewer #1: Yes

Reviewer #2: Yes

4. Is the manuscript presented in an intelligible fashion and written in standard English?

Reviewer #1: Yes

Reviewer #2: Yes

5. Review Comments to the Author

Reviewer #1: The authors of manuscript “Analysis of the mechanism of castor tolerance to Cd metal based on proteomics and metabolomics” reported that the physiological results mainly emphasize the super-sensitive responses of castor plant roots to Cd stress and the effects of Cd stress on plants’ antioxidant system, ATP synthesis and ion homeostasis. Nevertheless these points need to be addressed by the authors:

1. It seems better to replace common name of castor by its botanical name in the title

2. Please add some important data of current study in the abstract section

3. Please write the keywords alphabetically. Don’t write those words as a keyword which is part of manuscript’s title. The first letter of keyword shall be either capital or small, keep uniformity

4. L91-92: Write “C” of copper and “M” of mercury in small letter

5. L98-100 and 101: Please rewrite the sentence, make statement clear

6. Study following articles to improve this section:

https://doi.org/10.1016/j.ecoenv.2021.112047

https://doi.org/10.1016/j.chemosphere.2021.132332

https://doi.org/10.1016/j.scienta.2020.109203

https://doi.org/10.15244/pjoes/80806

7. L105: Castor oil extracted from the seeds is not toxic and has medicinal use

8. L141: Please write the complete term before using its abbreviation. Afterwards write abbreviation of that term concerned. However, do not write an abbreviation at the start of a sentence. Use this practice throughout the manuscript

9. Please inform about the physichochemical properties of sowing media

10. Did you sterilize the seeds before sowing?

11. What was pot size material and size? How much soil added per pot?

12. L145: Write “p” of plants in small letter

13. L161-169: Please rewrite the sentence, make statement clear

14. L216: Please maintain uniformity while giving heading/ subheading

15. L270: Please incorporate required correction

16. L272: Write “C” of castor in small letter

17. Please improve the discussion section, use latest references

18. Please delete unnecessary detail from the conclusion, write this section in one paragraph

Reviewer #2: Experimental work is ok. But there is language issue throughout the manuscript. In the main pdf file, comments have been provided to improve the manuscript. Please find comments in the main pdf file in attachment.

6. PLOS authors have the option to publish the peer review history of their article (what does this mean?). If published, this will include your full peer review and any attached files.

Reviewer #1: **Yes: **Nasim Ahmad Yasin

Reviewer #2: **Yes: **Imran Muhammad

---

## [Author Response · Author response to Decision Letter 0]

31 Oct 2022

Thank you for your comments on the revision of this manuscript. I have revised it one by one according to my comments. If the modification does not meet your requirements, I am willing to revise it again. As for the reply and revision of the manuscript, I have sorted it into a word document and returned it together with the manuscript

---

## [Decision Letter · Decision Letter 1]

26 Jan 2023

Analysis of the mechanism of Ricinus communis L. tolerance to Cd metal based on proteomics and metabolomics

PONE-D-22-20884R1

Dear Dr. Fenglan,

We’re pleased to inform you that your manuscript has been judged scientifically suitable for publication and will be formally accepted for publication once it meets all outstanding technical requirements.

Kind regards,

Ying Ma, Ph.D.

Academic Editor

PLOS ONE

Additional Editor Comments (optional):

Reviewers' comments:

Reviewer's Responses to Questions

**Comments to the Author**

1. If the authors have adequately addressed your comments raised in a previous round of review and you feel that this manuscript is now acceptable for publication, you may indicate that here to bypass the “Comments to the Author” section, enter your conflict of interest statement in the “Confidential to Editor” section, and submit your "Accept" recommendation.

Reviewer #1: All comments have been addressed

Reviewer #2: All comments have been addressed

2. Is the manuscript technically sound, and do the data support the conclusions?

Reviewer #1: Yes

Reviewer #2: Yes

3. Has the statistical analysis been performed appropriately and rigorously? 

Reviewer #1: Yes

Reviewer #2: Yes

4. Have the authors made all data underlying the findings in their manuscript fully available?

Reviewer #1: Yes

Reviewer #2: Yes

5. Is the manuscript presented in an intelligible fashion and written in standard English?

Reviewer #1: Yes

Reviewer #2: Yes

6. Review Comments to the Author

Reviewer #1: Authors of the manuscript "Analysis of the mechanism of Ricinus communis L. tolerance to Cd metal based on proteomics and metabolomics" have improved their manuscript.

Reviewer #2: The authors have addressed all the comments. I agree with acceptance of the revised version of the manuscript.

7. PLOS authors have the option to publish the peer review history of their article (what does this mean?). If published, this will include your full peer review and any attached files.

Reviewer #1: **Yes: **Nasim Ahmad Yasin

Reviewer #2: **Yes: **Muhammad Imran

---

## [Editor Report · Acceptance letter]

20 Feb 2023

PONE-D-22-20884R1 

Analysis of the mechanism of *Ricinus communis* L. tolerance to Cd metal based on proteomics and metabolomics 

Dear Dr. Fenglan:

I'm pleased to inform you that your manuscript has been deemed suitable for publication in PLOS ONE. Congratulations! Your manuscript is now with our production department. 

Kind regards, 

on behalf of

Dr. Ying Ma 

Academic Editor

PLOS ONE